# Chinese Consumers’ Attitudes and Potential Acceptance toward Artificial Meat

**DOI:** 10.3390/foods10020353

**Published:** 2021-02-07

**Authors:** Jingjing Liu, Élise Hocquette, Marie-Pierre Ellies-Oury, Sghaier Chriki, Jean-François Hocquette

**Affiliations:** 1INRAE, Clermont-Ferrand, VetAgro Sup, UMR1213, Recherches sur les Herbivores, 63122 Saint Genès Champanelle, France; marie-pierre.ellies@agro-bordeaux.fr; 2Université Catholique de Lyon, 69002 Lyon, France; elise.hocquette@gmail.com; 3Bordeaux Sciences Agro, CS 40201, 33175 Gradignan, France; 4Agroecology and Environment Unit, ISARA Agro School for Life, 23 rue Jean Baldassini, CEDEX 07, 69364 Lyon, France; schriki@isara.fr

**Keywords:** artificial meat, Chinese consumer, willingness to try, willingness to eat, willingness to pay, survey

## Abstract

The interest for artificial meat has recently expanded. However, from the literature, perception of artificial meat in China is not well known. A survey was thus carried out to investigate Chinese attitudes toward artificial meat. The answers of 4666 respondents concluded that 19.9% and 9.6% of them were definitely willing and unwilling to try artificial meat respectively, whereas 47.2% were not willing to eat it regularly, and 87.2% were willing to pay less for it compared to conventional meat. Finally, 52.9% of them will accept artificial meat as an alternative to conventional meat. Emotional resistance such as the perception of “absurdity or disgusting” would lead to no willingness to eat artificial meat regularly. The main concerns were related to safety and unnaturalness, but less to ethical and environmental issues as in Western countries. Nearly half of the respondents would like artificial meat to be safe, tasty, and nutritional. Whereas these expectations have low effects on willingness to try, they may induce consumers’ rejection to eat artificial meat regularly, underlying the weak relationship between wishes to try and to eat regularly. Thus, potential acceptance of artificial meat in China depends on Chinese catering culture, perception of food and traditional philosophy.

## 1. Introduction

“Conventional meat production systems” (which produce meat from farm animals that have been raised on farms or ranches), are facing important challenges such as environmental and animal welfare issues, food safety, public health, and the need to face the increasing worldwide population and associated protein demand [1,2]. It is claimed that the ever-increasing demand for meat is unlikely to be met through conventional meat production, due to the limited arable land and water resources [3]. As such, despite improvements of conventional breeding and production systems, researchers and private companies have devoted themselves to the development of better alternatives to meat, other than highly marketed vegetarian meat alternatives [4]. There is another type of meat alternative on the horizon–cell-based meat, or so-called artificial meat, also known as cultured meat, cultivated meat, in vitro meat, or lab-grown meat [5,6], which can be grown from live animal stem cells rather than from farm animals [7]. Based on the hypothesis that artificial meat may be able to alleviate the aforementioned ethical, environmental, health, and hunger problems resulting from conventional meat production [8], artificial meat is currently a hot topic within the industrial, political, societal, and scientific areas in both the scientific field and the press media [9,10]. In addition to the hypothesis that artificial meat is presented by the press media to be able to address the aforementioned problems in Western countries [10], it could also mitigate the same major issues for China. As early as 2007, global food demand is expected to be 77% higher by 2050, China is driving this demand by accounting for 43% of the global agri-food increase [11]. Meat demand in China is anticipated to grow a lot from 2019 to 2050, given the sustainable growth of China’s middle class, the meat demand would be higher than nowadays [12]. In particular, beef consumption in China from 2015 to 2050 is expected to increase by 119% [13]. However, meat production in China is increasing at a lower rate than the demand for meat [14]. Meat shortage may be a major sustainable development challenge that China must face in the future, which means they need to continue exploring and adopting effective alternatives to fill this vacancy, such as artificial meat produced from cell culture.

Up to the present time, there have been various surveys regarding consumer acceptance of artificial meat, with most research focusing on Western countries [15,16], such as the United States and European countries. These studies found that more than half of the respondents who participated in these surveys would be willing to try artificial meat but fewer participants would be willing to eat regularly or pay for this novel product [17]. In fact, due to the cross-cultural differences in consumer acceptance of artificial meat, these results cannot be extrapolated from Western countries to Chinese consumers [8]. Many consumer-facing companies now realize that they will need to penetrate the Chinese market to boost their development [18]. Scientists also speculated that this novel food product, advertised as clean, (i.e., wholesome, sustainable and less polluting, since potentially using fewer hormones, less resources, and being less harmful for the environment), could be the silver bullet for China [19]. However, limited research has explored consumer acceptance of artificial meat in China [20]. Nevertheless, Chinese consumers’ opinions and the potential Chinese market have been already studied by the Cellular Agriculture Society [19]. A high potential acceptance of artificial meat has been observed but another study indicated that approximately 50% of Chinese consumers have no opinion [21]. Other authors who conducted a survey in three countries (the USA, India, and China) confessed the possibility that some Chinese participants did not fully understand the concept of artificial meat which might be confounded with plant-based meat [20]. Therefore, more reliable data and new investigations are needed to investigate artificial meat acceptance and potential demand of Chinese consumers for artificial meat with more assurance that the concept will be fully understood. In addition, strong evidence is still lacking about the perception by Chinese consumers how artificial meat may contribute to solving the above-mentioned problems related to food security, animal welfare, and environmental protection. In light of its distinctive catering culture, view of food, and traditional philosophy, Chinese attitudes toward artificial meat need multi-aspect and specific exploration.

Therefore, the current study seeks to work with a large group of Chinese consumers who are likely to better understand the concept (compared to previous studies) and with appropriate questions to deeper investigate their attitudes, perspective, potential acceptance, and willingness to engage with artificial meat, and to provide a Chinese reference to the widest artificial meat market worldwide. Some of the major potential predictors of willingness to try, eat regularly, and pay for artificial meat, as well as potential reasons for willingness or non-willingness to try and perception about the feasibility of this novel product have been also explored.

## 2. Materials and Methods

### 2.1. Data Collection

An online survey entitled “Survey on novel food—Artificial Meat” was conducted in China between 15 June and 30 August 2020. A total of 4929 survey responses were collected by different means including the dissemination in universities and commercial providers of sampling services. The process adhered entirely to the ESOMAR (European Society for Opinion and Market Research) guidelines on ethical online research [22]. This includes assurances that respondents gave informed explicit consent to take part in the survey and had their personal data protected. Indeed, after being informed of the objectives of the survey and how the given information will be used, all respondents gave their informed consent for inclusion of their answers before and after they participated in the study. Respondent details (sex, age, education level, activity area, monthly net income, meat consumption, and familiarity of respondents with artificial meat) have been collected in an anonymous way with an option “no wish to answer” and with no personally identifiable information. Otherwise, this work is part of an international survey (available in different languages) conducted following local guidelines based on the laws and regulations of the countries in which the research has been performed (in this case, based on practices of China agricultural University [21]) and this includes ethical approval by ethics committees when required (such as in Brazil: number CAAE: 37924620.5.0000.5404).

Responses when total survey duration was less than half of the median duration were excluded to avoid irrelevant answers to some extent. Indeed, these answers might be less accurate and therefore could cause biased results. The final sample consisted of 4666 participants, which means a confidence interval of 1.43% maximum based on the size of the Chinese population and depending on the different proportions of answers to the different questions (https://www.surveysystem.com/sscalc.htm (accessed on 6 February 2021)).

### 2.2. Questionnaire Design

The survey consists of four sections. Before starting to answer the questionnaire, the introduction of artificial meat in the form of text and graph was briefly provided to respondents (Figure 1). Unlike previous studies conducted in China, a picture describing the principles of artificial meat production has been provided in addition to a small text to avoid as much as possible any misunderstanding or any confusion with other meat substitutes as suspected in previous surveys [20,21]. Indeed, a pre-test of the survey showed that the respondents were indeed sometimes confused between plant-based meat and artificial meat produced from cultured cells. The survey was thus modified after discussion with these first respondents in order to consider the psychology of Chinese respondents. Indeed, this combination of quantitative and qualitative data as recommended in psychometrics [23] was crucial to minimize any confusion. As a result, the explanatory figure was added and invalid responses were removed.

After being informed, the respondents could answer the survey according to various criteria. Sociodemographic information was collected in the first section, including sex, age, education level, activity area, monthly net income, meat consumption, and familiarity of respondents with artificial meat. Then, one question was asked as preamble regarding respondents’ food purchase criteria. The second section provided information about respondents’ attitudes toward societal challenges faced by conventional meat and artificial meat, regarding ethical, environmental issues, the traditional meat industry, and rural life issues. Afterwards, the personal perceptions about artificial meat compared to conventional meat or plant-based meat were asked within the third section. After these sections, the respondents were asked about their willingness to engage with artificial meat with respect to willingness to try (WTT), how much they would be willing to pay (WTP) and under which context they would be willing to eat or not artificial meat regularly (WTE). Finally, future development strategies were asked concerning the respondents’ opinions towards whether artificial meat is realistic, as well as the feasibility of the development of artificial meat by private and public research models. The survey is detailed in Appendix A.

Artificial meat, also known as cultured meat, in vitro meat, cultivated meat, lab meat, clean meat and synthetic meat, is a novel food produced in laboratories using animal muscle stem cells, but does not come directly from a living animal and which proliferate in culture. The production of artificial meat is the subject of media enthusiasm to feed the growing human population. In order to address the increasing concerns about environment (global warming) and ethic (animal welfare) but also the weakness of the conventional meat production (limited farming resources and low efficiency to feed the ever increasing population), scientific research is devoted to introduce and develop on a large scale artificial meat as a novel food in the future.

### 2.3. Statistical Analysis

Data were analyzed by using R software (version.3.5.2) [24] and IBM SPSS 25 [25] using a variety of statistical techniques as previously done [26]. As the first step, the distribution of answers was summarized based on number of responses, percentage of responses, mean value, and standard deviation (SD) by R function “describe.”

In order to visually present the responses of multiple-choice questions, the word cloud was used to provide the overview by distilling the size of text based on words that appear with frequency [27] by “wordcloud” package and “comparison.cloud” function. The word cloud graph was generated from three multiple-choice questions, which were: (1) What could be the reasons for you to have willingness to try artificial meat? (2) What could be the reasons for you to have no willingness to try artificial meat? (3) What would you expect from artificial meat?

As in previous studies [20], some, but not all of the assumptions of ANOVA were sometimes violated in this survey case, such as normality of distributions and homogeneity of variances. Therefore, we ran a Welch’s ANOVA, which does not require the homogeneity of variance assumption, and we obtained extremely similar results compared to ANOVA which is considered as being robust [28]. Based on these observations, we proceeded with ANOVA since the Welch’s ANOVA does not accept interactions. Variance analysis was performed in SPSS to determine whether the demographic variables affect respondents’ willingness to try, to eat regularly, and finally to pay. The main effect of each of the demographic factors on the willingness to try, to eat, and to pay was first studied to select the significant factors by Welch’s ANOVA, which were further used to study the interaction effects by ANOVA based on the robustness of the ANOVA under the application of both normally and less-normally distributed data [28]. Then the post-hoc test was performed by using a Tukey HSD test with the Bonferroni correction for the pairwise comparisons between significant groups. Differences were considered significant at a Bonferroni-corrected *p* < 0.05. The significant differences were confirmed by other statistical approaches such as the Student *t*-test adapted for heterogenous variances and also non-parametric tests. In addition, the responses regarding willingness to pay and “age × familiarity” was performed in mosaic plot by R package “vcd” and “mosaic”.

With a large amount of qualitative data, it is relevant to use some exploratory techniques to explore the underlying relationship structures among multiple, diverse and categorical variables. Thus, multiple correspondence analysis (MCA) was chosen and applied in this study to detect and reveal the exploratory insights among multiple aspects of consumer behavior, intention, and willingness. Particularly, MCA can be used to represent and model datasets as clouds of points in a multidimensional space, showing if relationships exist between variables and offering statistical results that can be seen both analytically and visually [29]. The MCA was performed by R package “FactoMineR” and “MCA” functions.

In addition, cluster analysis defines and classifies homogeneous variables into groups based on the MCA dimensions assuming that they have substantive coherence [29]. The principle of clustering is by default based on the Euclidean distance, which is the square root of the sum of the square differences. In this study, data for willingness to try and willingness to eat regularly [Question1: would you be willing to try artificial meat?, divided into two groups: No Willing to try (score of 1 and 2, when participants answer 1-definitely no willing to try or 2-probably no willing to try, which were both combined as no willing to try in MCA analysis) and Willing to try (score of 4 and 5: 4-probably willing to try or 5-definitely willing to try); Question2: would you be willing to eat artificial meat regularly?, divided into two groups: Willing to eat regularly; No Willing to eat regularly], were then analyzed to select the categorical variables with similar characteristics, regarding societal challenges and respondents perception of artificial meat. According to the results of MCA, the first two dimensions (for both WTT and WTE) were chosen to calculate the Euclidean distance from each variable to the two categorical groups, and the variable with the smallest distance to each center group was allocated to the corresponding cluster.

## 3. Results

### 3.1. Distribution of Answers

#### 3.1.1. Demographic Information of the Respondents

According to the sociodemographic information detailed in Table 1, the current sample of respondents was characterized by quite equally females (50.6%) and males (46.2%) who were mainly young and middle-aged people (77.6%), no scientist and outside the meat sector (74.3%), medium education level [(college, 55.6%; master, 20.4%)], medium income (3000–8000 yuan, i.e., around 460–1240 USD or 380–1020 Euros, 38.1%) and meat-eaten people (regularly and daily meat eater, 73%). Among them, 55.9% of respondents have heard about artificial meat before.

#### 3.1.2. Societal Challenges

As shown in Table 2A, nearly half of the respondents agreed with the potential ethical and environmental problems that may be caused by conventional meat production (47.4 and 50.0% respectively for the sum of scores 4 and 5), and 51% of respondents believed reducing meat consumption could be a good solution to solve the potential ethical and environmental problems caused by conventional meat production. Around half of the respondents thought that artificial meat is more ethical and eco-friendly than conventional meat. However, more than 50% of the respondents admitted that artificial meat would have negative impacts on the traditional meat industry, territories, and rural life. Nonetheless, for all these questions, many respondents were unsure (i.e., from 23.3 to 33.6% of score 3, Table 2A).

#### 3.1.3. Perception

More than half of the respondents (sum of scores 1 and 2) believed that artificial meat will be less healthy, less safe, less tasty, and with a lower nutritional value than conventional meat (Table 2B). A total of 49% of respondents thought this novel food is fun and/or intriguing, 36% thought artificial meat is promising and/or acceptable, and 15% of the respondents indicated that artificial meat is absurd and/or disgusting (Figure 2). Relative to 16.1% of the respondents who had high emotional resistance to try artificial meat (sum of scores 4 and 5), 52.3% had less emotional resistance to try (scores 1 and 2) whereas 31.5% of them were unsure (Table 2B).

#### 3.1.4. Willingness to Engage

A total of 49.7% of respondents would be willing to try artificial meat (29.8% probably yes, 19.9% definitely yes), 20.9% respondents expressed their unwillingness to try artificial meat (11.3% probably no, 9.6% definitely no), and 29.3% were unsure about their willingness to try it (Table 3A).

A great part of the respondents (70.1%) already eat meat substitutes. Around 52.9% of the respondents (irrespectively of the fact they already eat meat substitutes or not) will accept artificial meat as a good alternative to conventional meat (Table 3A). However, it is noteworthy that 34.1% of the respondents who already eat meat substitutes (in the form of plant-products) will refuse to accept artificial meat as an alternative to conventional meat. Finally, 12.9% do not accept artificial meat nor plant-based meat substitutes (Table 3A).

Of the total respondents, 47.2% do not want to eat artificial meat regularly at all. By contrast, 39.7%, 35.5%, and 35.6% of them are willing to eat artificial meat regularly at restaurants, home, and/or in ready-to-eat meals respectively.

About 46% of respondents are willing to pay for artificial meat at a much cheaper price than conventional meat (even nothing at all), 40% are willing to pay at a lower price than conventional meat, 10% are willing to pay at the same price as conventional meat. Only 3% of respondents are willing to pay more (data shown in Table 3A).

#### 3.1.5. Reasons for Willingness or No Willingness to Try and for Expectations about Artificial Meat

Globally, 74.2% of respondents agreed to try at least once artificial meat but 25.8% answered they do not want to (data not shown). Food crisis (data shown in the note of Figure 3), avoiding zoonosis, and curiosity are presented as the first three important reasons associated with willingness to try, but ethical and eco-friendly issues are not the main reasons compared to the concerns of food shortage and security. There were a number of people (22.9%) who had indicated that they had no willingness to try artificial meat.

Reasons for willingness to try: food crisis: 31.3%; Less risk of zoonosis (disease that can be transmitted from animals to people): 30.9%; Curiosity: 26.8%; Ethics (improve animal welfare and reduce animal slaughtering): 22.7%; Attractive price compared to conventional meat: 21.4%; Clean product: 21.2%; Eco-friendly product: 20.1%; high-tech product: 19.7%.Reasons for no willingness to try: unsafety: 42.5%; unnaturalness: 34.1%; distrust in lab-produced product: 26%; emotional resistance (disgust, nervousness): 25.1%; no appealing/tasty: 23.9%; expensive: 19.8%; negative impact on farming: 16.3%; negative impact on territories: 14.5%; environmental footprints: 13.8%.Expectations for artificial meat: safety: 47.5%; being tasty: 42.9%; adequate nutrition: 42.4%; less carbon footprint: 34.9%; cheap: 34.4%; no animal suffering: 29.8%; reduce farming: 19.9%; no farming: 15.6%; nothing expected: 9%.

Unnaturalness and unsafety were frequently chosen as the main reasons for no willingness to try whereas environmental footprints and negative impact on territories were the last two reasons for their no willingness to try (Figure 3).

In terms of expectations about artificial meat, safety, taste, and adequate nutrition were the most often selected (47.5; 42.9, and 42.4% respectively) compared to other options such as reducing or no farming (19.9 and 15.6% respectively). Many respondents expected also artificial meat could solve the problems of carbon footprints and animal suffering (42.9 and 34.4% respectively). Nearly 9% of respondents answered having no expectations from artificial meat (Figure 3).

#### 3.1.6. Future Development Strategies

Respondents were asked about their opinion on development of artificial meat. For 45.1% of respondents, the development of artificial meat will become realistic in the medium term (i.e., from 6 to 15 years) but for 10.3% of respondents, artificial meat will never be realistic (Table 3B).

About 46% of respondents answered that the public research should invest much more or more time and funding to develop artificial meat, 14.5% of respondents thought that public research should invest much less or less in this biotechnology development. In contrast, 51.8% of respondents considered that private research models are much less or less relevant and only 14.6% thought start-ups are relevant for artificial meat research development (Table 3B). In both cases, between 33.5% and 39.5% are unsure about these aspects.

### 3.2. Effects of Demographic Factors on Willingness to Try, to Eat Regularly or to Pay for Artificial Meat

In order to perform further analysis, representativeness of the current sample regarding sex and age was analyzed. There were 50.6% females and 46.2% males in this study, the gender ratio for China population is 48.69% females and 51.09% males. Whereas in the current study, respondents were on average young (with 46.1%, 31.5%, and 22.4% of respondents between 18 and 30 years of age, 31 and 50 years, and more than 51 years, respectively), the corresponding age structure of China population is 23.4%, 38%, and 38.6% for the same age groups [30]. Despite the fact our sample of Chinese consumers seems to be non-representative particularly based on age, the current demographics were studied as important potential factors to explain willingness to try (Table 4 and Table 5) or to regularly eat cultured meat and to pay for artificial meat (Table 6).

#### 3.2.1. Relationships between Demographic Factors and Willingness to Try

Willingness to try artificial meat is regulated by all the studied demographic factors (sex, age, education level, activity area, income, level of meat consumption, and familiarity with artificial meat), which act together in interactions (Table 4).

As it is shown in Table 5A, for the current participants, men over 31 or 51 years of age were more willing to try artificial meat than women of the same age. Women are characterized by an increasing willingness to try with increasing age, whereas for men, willingness to try increases between 18–30 and 31–50 years of age only.

People who had a master’s degree (especially men) have more willingness to try artificial meat than other participants with different levels of education (Table 5B).

Participants who perform an activity within the meat sector have the highest willingness to try artificial meat, especially female meat workers. The group with the lowest willingness to try is females who are not scientists and who work outside the meat sector. On average, scientists have greater willingness to try than other female participants outside the meat sector (Table 5C).

Males with medium income have the highest willingness to try and females with the lowest income have the lowest willingness to try (Table 5D).

Males who are high meat-eaters would be more willing to try artificial meat followed by females of the same level of meat consumption. On the opposite, males, and to a lower extent of females, who never eat meat, have the lowest willingness to try (Table 5E).

Whereas willingness to try generally increases with increasing age, old participants with the highest education level have more willingness to try than younger and low educated participants. Moreover, young but high educated respondents are less willing to try artificial meat to a large extent (Table 5F).

The group of old people has not always the highest willingness to try artificial meat since old participants who never eat meat have the lowest willingness to try even lower than young and mid-aged vegans. By contrast, old meat daily eaters would be the best volunteers to try artificial meat compared to younger participants or less regularly meat-eaters (Table 5G).

#### 3.2.2. Relationships between Effective Demographics and Willingness to Eat Regularly and Willingness to Pay

On average, willingness to eat regularly and willingness to pay do not depend too much on demographic factors compared to willingness to try (only significant interactions are shown in Table 6). In line with previous literature underlying the importance of familiarity for consumers’ acceptance of this novel product, the “age × familiarity” interaction for willingness to pay is significant (Table 6). The relationships among willingness to pay, familiarity, and age are therefore investigated, and results presented in a mosaic plot (Figure 4).

As shown in Figure 4, young respondents who already heard about artificial meat would be willing to pay less than for conventional meat. Furthermore, people who are older than 31 years of age and never heard about artificial meat before are likely to pay less for artificial meat or the same price compared to conventional meat. A very small proportion of middle-aged people who are unfamiliar with artificial meat are inclined to pay more than for conventional meat.

Positive and great values of Pearson standardized residuals are observed in cells colored in blue. Negative and great values of Pearson standardized residuals are observed in cells colored in red. Standardized residuals are standardized differences between observed and theoretical effectives. There are associations between respondents’ age, familiarity, and their WTP for artificial meat that can be observed. These results were confirmed by significant differences between mean values of WTP following variance analysis: 18–30/No (1.89 ^a^) > 31–50/Yes (1.74 ^b^); 18–30/Yes (1.73 ^b^); 51-/No (1.7 ^b^); 31–50/No (1.7 ^b^) > 51-/Yes (1.59 ^c^) where 1 corresponds to the answer “WTP less (or much less)”, 2 to the answer “WTP the same as conventional meat”, 3 to the answer “WTP more (or much more)” than for conventional meat.

### 3.3. The Driven Factors of Willingness to Engage with Artificial Meat

It can be seen from the MCA plot of societal challenges that respondents can be distinguished according to their score of willingness to try and to eat artificial meat regularly (Figure 5a,b).

Respondents with willingness to try (Figure 5a) and to eat regularly (Figure 5b) answered to a larger extent that “conventional meat cause ethical and environmental problems”, “artificial meat is ethical” and “reduce meat consumption could be a good solution”. They also answered that “artificial meat has a negative impact on the traditional meat industry, territories, and rural life.” On the other hand, disagreements with the same statements (i.e., “conventional meat cause ethical and environmental problems”, “artificial meat is ethical”, and “reduce meat consumption could be a good solution”) are associated with no willingness to try and to eat regularly.

Respondents who believe artificial meat can be very safe, healthy, nutritional and tasty compared to conventional meat would be willing to try (Figure 6a) and to eat this novel food regularly (Figure 6b). On the contrary, respondents have definitely no willingness to eat regularly when they believe artificial meat products will be much less safe, healthy, nutritional and tasty compared to conventional meat. However, when participants are concerned by safety, healthiness, as well as by nutritional and sensory traits of artificial meat, they still would be willing to try. Responses such as “no emotional resistance to artificial meat” are related to willingness to try and to eat regularly.

It is obvious that respondents who have extremely emotional resistance to artificial meat tend to perceive artificial meat as “absurd and/or disgusting”, which would lead to no willingness to try and to eat regularly (Figure 6a,b). The response of “artificial meat is “promising and/or acceptable” is related to the willingness to try and to eat regularly clusters, whereas “artificial meat is fun and/or intriguing” is located both at “willing to eat regularly” and “no willing to eat regularly” clusters (Figure 6b). Respondents who already eat meat substitutes and will accept artificial meat as alternatives would be highly willing to try and eat regularly. However, respondents who do not eat meat substitutes but somehow accept artificial meat as alternatives would be willing to try and to eat regularly. People who do not accept artificial meat as alternatives would be unwilling to eat artificial meat regularly (Figure 6b).

It is presented in Figure 6a that “ever heard” and “never heard” about artificial meat are both associated with willingness to try. By contrast, “ever heard” about artificial meat is related to both willingness or no willingness to eat regularly, “never heard” about this novel food is related only with no willingness to eat regularly (Figure 6b).

In addition, in order to further confirm the relationships between respondents’ perspective and willingness to engage with artificial meat, the three variables of interest (willingness to try, to eat regularly, and to pay) are all presented in Figure 6b. Willingness to eat regularly is close to “artificial meat is promising and/or acceptable” and “I accept artificial meat as an alternative and I eat meat substitutes.” Willingness to try is close to “I do not have emotional resistance to artificial meat.” Willingness to pay the same as conventional meat or less” is close to “artificial meat is fun and/or intriguing.” No willingness to try and to eat regularly are both close to “artificial meat is absurd and/or disgusting.” No willingness to try is closer to “I have emotional resistance to artificial meat”. The answer “I do not accept artificial meat as an alternative and I do not eat meat substitutes” is closely related to “artificial meat is absurd and/or disgusting.”

### 3.4. Relationships among Respondents’ Attitudes, Perspective, Acceptance and Willingness to Try and to Eat Regularly

Respondents were asked to indicate their most important criteria during food purchase from ten options: safety, sensorial quality, and price are ranked as the top three food purchase criteria. Environmental impact and ethics may not be the priority considerations for most of the current respondents. The cross-analysis between food purchase criteria and willingness to try artificial meat is presented in Figure 7. People who want the least to try artificial meat are those with the highest concern for nutritional value, safety, and traceability. By contrast, people who would be relatively more willing to try are respondents who pay the highest attention to labels, energy intake, and ethics.

Figure 8 indicates a positive association between willingness to try and willingness to eat regularly artificial meat. However, this cross-analysis also indicates that some respondents with no willingness to eat artificial meat regularly may agree or probably agree to try once. However, 11% of respondents who would be willing to eat artificial meat regularly at a restaurant, home, or in ready-to eat meals, have no willingness to try artificial meat before regular consumption, which may suggest they already trust the food-tech industry.

As shown in Figure 9, respondents who will accept artificial meat as an alternative to conventional meat generally have a high willingness to try (63%) and a high willingness to eat regularly (62%), which is logical. Among them, a great part of them already eat meat substitutes (52% and 48% for willingness to try or eat regularly respectively). In other words, for respondents who would be willing to try artificial meat and would accept it as a viable alternative to conventional meat compared to meat substitutes, 52% of them already eat meat substitutes and 11% have not eaten meat substitutes but would accept artificial meat as an alternative.

For people who had no willingness to try artificial meat, 70% of them would not accept artificial meat as an alternative. Nearly 62% of respondents who would be willing to eat artificial meat regularly would accept it as a viable alternative to conventional meat, including 14% of them who do not eat meat substitutes.

Around 38% of respondents who would be willing to eat artificial meat regularly would not accept it as an alternative to conventional meat. On the opposite, 41% of respondents, who had no willingness to eat artificial meat, would accept artificial meat as a viable alternative to conventional meat.

The observations between respondents’ perspective of artificial meat and their willingness to eat regularly is presented in Figure 10. Nearly one-quarter of people who have no willingness to eat artificial meat regularly find artificial meat is “absurd and/or disgusting” whereas, surprisingly, about one quarter find it “promising and/or accepting.” For half of the respondents who have no willingness to eat regularly, artificial meat is “fun and/or intriguing.” By contrast, for people who would be willing to eat regularly at a restaurant, home or in ready-to-eat meals, a large majority of them thinks this novel product is “promising and/or accepting” or “fun and/or intriguing.” However, there is still a small portion of people who find artificial meat is absurd and/or disgusting but still would be willing to eat it regularly.

## 4. Discussion

### 4.1. Chinese Consumers Are Not Eager for Artificial Meat

Bryant et al. [20] observed that Chinese consumers’ attitudes toward artificial meat were more positive than consumers in the USA. However, Siegrist and Christina [31] found that American consumers did not differ from Chinese consumers in accepting artificial meat. In addition, Zhang et al. [21] found that nearly 22% of Chinese participants were opposed to artificial meat, and approximately 50% remained neutral. Consequently, only 28% expressed their positive attitude in favor of artificial meat, which reflects that the Chinese consumers’ attitudes toward artificial meat are quite conservative. It was also demonstrated that Chinese consumers are among the least likely to eat artificial meat (26%) compared to some other Asian countries (Vietnam, Thailand, Indonesia) [32]. In brief, the current literature about potential acceptance of artificial meat in China did not reach so far consensus on this subject. In the current survey with 4666 Chinese consumers, we found that only a fifth of respondents were definitely willing to try, nearly half of them were not willing to eat regularly, and almost 90% were willing to pay for artificial meat at a lower price than conventional meat. Obviously, these findings cannot be interpreted as Chinese consumers are eager for artificial meat.

Anderson and Tyler [33] found that Chinese respondents were more likely to provide responses in the middle of the scales than respondents in the United States. This is consistent with our study indicating that 30% of respondents stated that they are “unsure” or “neutral” and about 40% of respondents may or may not agree, perhaps with a yes or no, to nearly every question. On one hand, people are likely to be truly unaware of the related issues such as the societal concerns facing traditional meat that are likely to be resolved by artificial meat. On the other hand, this may be due to the influence of “golden mean”, which has always been the basic concept of Chinese philosophy. People in different countries answer survey questions with different habits. However, a too extreme response does not always reflect their sense of worth. Especially for the Chinese consumers, who prefer any implicit expression. In other words, the uncertain answers such as “unsure” or “probably” are more likely to be related to the rejection.

In light of the large numbers of consumers and relatively permissive regulatory frameworks, it has been hypothesized that Asia is currently the first consumer market for artificial meat [34]. However, since the Chinese market of artificial meat has not been deeply explored despite its growing interest in Western countries, Chinese philosophy (e.g., the way of thinking as well as expression) should be taken into account.

### 4.2. What Kind of Consumer Would Be Willing to Engage with Artificial Meat in China?

#### 4.2.1. Artificial Meat Is More Attractive to the Elderly and to Men in China

Some previous studies have observed, in different countries, that artificial meat is more attractive for young people and men than for older people and women [16,35]. This is understandable because new innovative things are more likely to be accessible and acceptable by young people compared to old ones. However, in contrast, the current study found that middle-aged adults and old people (over 31 years of age) were more willing to try artificial meat than young people. In line with previous conclusions, Chinese men were indeed more willing to try artificial meat than Chinese women, and young Chinese had a more conservative attitude towards artificial meat. Regardless of the fact that the current samples of respondents were not sufficiently representative of the Chinese population based on age, the comparisons between age groups were still relevant in this study. These differences between age groups may be due to different consumption concepts based on different economic strengths. With the accumulation of economic and social resources by middle-aged and elderly people, the latter became more curious and open-minded to accept novel things, to compensate for the lack of impulse purchases in their youth due to a very low economic activity in China several decades ago. By contrast, Chinese young people, who are more beginners in being economically independent, being still students, with economic and parental pressure, may tend to be more conservative face to new things, especially if the price of the new product is too high. Nevertheless, the artificial meat market could be developed not only for middle-aged adults but also for young people due to the increasing tendency of modern eating habits based on convenient snacking. This modern habit can be easily explained by the busy work schedules of young and especially middle-aged people in China [36].

#### 4.2.2. Artificial Meat Is More Attractive for Daily and Regularly Meat Eaters

Meat eaters are found more willing to engage with artificial meat than vegetarians [37]. Our study confirmed that daily meat eaters had a higher willingness to try artificial meat than regularly and rarely meat-eaters. Despite the consideration that artificial meat can be regarded as a vegetarian product, which could be attractive to some vegetarians and vegans [38], it was well confirmed by several studies that artificial meat is more appealing to meat-eaters than vegetarians [39,40]. Our study confirms that vegans and/or vegetarians were more unwilling to try artificial meat expectedly. For the current respondents, there are only 3.4% vegetarians, which indicates that the majority of meat-eaters are likely to be considered as the potential target for the acceptance of artificial meat. In addition, during the COVID-19 pandemic, due to the increased weaknesses of the global food supply chain, some people are now expecting the transition to artificial meat to happen even faster [41] and this may have influenced answers to this survey towards a great acceptance of artificial meat.

#### 4.2.3. Are Education and Activity Area Relevant for Willingness to Try Artificial Meat?

It was demonstrated that highly educated people are more inclined to hold analytic, deliberative standpoint rather than based on heuristics and biases which may be more likely to lead to higher acceptance of artificial meat [8]. Nevertheless, Hocquette et al. [42] found that educated people were not convinced by the fact that artificial meat is the solution to the problems of the conventional meat industry. For the current participants, willingness to try is poorly associated with educational level. To sum up, based on the current sample, people with master’s degree might be more willing to try, this might be due to their generally scientific background and their level of knowledge may stimulate their acceptance of new tech and concept. Education background might contribute to willingness to engage with artificial meat since people with doctorate degree are more likely to be rigorous to strictly refuse or accept artificial meat. People who were less educated may also be very open or stubborn toward this novel product. However, it is still relevant to emphasize the slightly non-representativeness of the current data for the entire Chinese population although comparisons between social groups are still valid. Indeed, as stated by Heidemann et al. [43], respondents from different regions could provide new multiple perceptions and enrich the existing knowledge even if they are non-representative of the whole country. Furthermore, we observed that highly educated respondents were poorly aware of this novel product [40]. Further research is therefore needed based on the education perspective.

It was found that cultured meat acceptance is significantly higher amongst agricultural and meat workers, indicating that those who are closest to existing meat production methods are most likely to prefer alternatives [26]. The present study also observed that participants who work in an area related to the meat industry had a higher willingness to try artificial meat, this may be explained by the fact that people working related to meat production are more curious about the novel product itself and willing to have a first try at least. However, it does not mean they would be willing to eat regularly or to pay at an expensive price. Since no significant effects were observed in variance analysis between activity area and willingness to eat and pay, the conclusion of Bryant et al. [26] was not confirmed in our study.

#### 4.2.4. Unfamiliarity with Artificial Meat May Lead to a First Try but Not Likely to Eat Regularly and to Pay More

People familiar with the conception of artificial meat are more likely to have higher acceptance according to previous studies [8]. Indeed, Rolland et al. [44] concluded that having positive information improves acceptance and willingness to taste artificial meat. For the current Chinese respondents, with quite neutral information provided, over half were familiar with artificial meat. In contrast to another survey conducted in China in which the term “cultured meat” was used, most people declared that they had never heard of cultured meat or in vitro meat [21]. This discrepancy may be explained by familiarity with the different terms or names used instead of familiarity with the novel product by itself [8]. Consequently, previous results might be biased due to some confounding effects of different naming. For instance, the Chinese translation of artificial meat is “man-made meat”, which might be confounded with plant-based meat; cultured meat is more likely to be known or perceived as “farmed/nurtured meat” and cultivated meat can be translated into “cultivated and reproduced meat”, which are more unfamiliar terms to Chinese people compared with artificial meat.

We observed that people who have never heard about artificial meat would be willing to try and pay less or pay the same price as conventional meat likely due to the curiosity for this novel technology. However, for those who never heard about artificial meat, they had no willingness to eat regularly. The finding that people not yet aware of this novel product would be willing to try and even pay for it but unwilling to eat regularly indicates that regular engagement is unacceptable for them at the present stage.

On the other hand, when consumers fully understand the concept of the new product, some of them refuse or are unsure to try at the very early beginning maybe because of some reasons such as food neophobia and emotional resistance. However, others would be willing to pay for artificial meat but with a lower price (as for young participants) probably because, from their perspective, artificial meat will be never better than real meat. This might reflect a special pattern of Chinese consumers who have a lack of knowledge about this novel product. Whereas it has been suggested that the limited knowledge curbs the development of artificial meat [45], by contrast, the core value of food products lies not only in physical aspects but also in long-term added value [46]. For instance, willingness to pay about animal welfare in relation to a social consensus has clearly a moral value [47]. When informing consumers about environmental or animal welfare benefits, their willingness to consume increases [16,48]. Herein, propagating the knowledge about positive impacts on animals provided by artificial meat can be regarded as an effective way to improve consumer acceptance in China.

### 4.3. Potential Changes of Drivers of Chinese Consumers’ Willingness to Try or to Eat Artificial Meat

Several factors such as unnaturalness, food neophobia, food disgust, price, and taste were identified as major factors influencing consumer acceptance of artificial meat in Western countries [48,49,50]. A substantial cultural difference between China and other countries may cause different inducement of accepting or rejecting artificial meat [20]. Regarding global hot issues of social concern such as environment protection, animal welfare, and world hunger which are anticipated to be alleviated by artificial meat [8], the views of most Chinese respondents might be consistent to some extent. The questions related to these issues were therefore asked to address societal challenges, though the comparison of artificial meat and conventional meat.

#### 4.3.1. Ethical and Environmental Issues Can Be Drivers of Artificial Meat

The societal challenges considered in this study were expressed in terms of ethical and environmental problems associated with meat production, in favor of meat consumption reduction (even veganism) or in relation to traditional issues. It was found that, for participants who believed that conventional meat causes ethical and environmental problems, artificial meat appears more ethical and eco-friendly compared to conventional meat. For these respondents, reducing meat consumption is a good solution to resolve the above issues, and they would be willing to become vegetarian and/or to try artificial meat with a high probability. This indicates that people with a more favorable perception of artificial meat might be more knowledgeable on this subject related to ethics and environmental issues with a more unfavorable perception of the conventional production system.

Despite the fact that these respondents agree with the idea that artificial meat will have negatively impacts on traditional meat industry, territory and rural life, they still expressed their willingness to try. Consequently, in line with our previous discussion, it might be hypothesized that an effective way to improve consumer acceptance of artificial meat might be to focus on the problems of conventional meat production, to emphasize the potential benefits of artificial meat, and explain how artificial meat can be substituted to conventional meat production. It is important to note that when respondents were asked about their expectations on artificial meat, nearly 30% expected reducing farming and 20% even hope for “no farming.” It seems that despite the long history of agriculture, some Chinese people do not place considerable importance on the traditional agricultural industry.

#### 4.3.2. Most Consumers Are Unsure and Have Less Strong Opinions about Artificial Meat

It was previously reported that Chinese consumers only focus on their personal reasons to eat or not artificial meat [51]. The current participants indeed expected positive internal characteristics (e.g., sensorial quality) of artificial meat. However, willingness to try or not to try might not be associated with any negative or positive characteristics of artificial meat products, since the responses about the characteristics “artificial meat is less tasty” and “artificial meat is less safe, healthy and nutritional” are all related to willingness to try. The first engagement with artificial meat might be less strict since only a first try would probably not lead to negative consequences. However, this is not the same case for willingness to eat regularly. When respondents were asked about their willingness to eat, their concerns about taste, safe, healthiness, and nutrition became more important. For those who think artificial meat has fewer advantages in the aforementioned characteristics, they would refuse to eat it regularly.

Extremely strict boundaries may not always exist between acceptance and rejection. For some swing consumers, the internalities of artificial meat such as taste and appearance might not so important. Consumer behaviors are complex and multifactorial. The conclusion that can be drawn from this study might be that the large number of swing consumers are the real market to look for. Moreover, the producers may seize the chance of consumers’ willingness to have a first try, to convince their potential consumers by resolving the concerns about safety and the others. Besides, extremely emotional resistance to artificial meat in line with the perspective of “artificial meat is absurd and/or disgusting” will lead to the rejection of artificial meat directly. However, nearly half of respondents who think “artificial meat is fun and/or intriguing” would probably willing to try and pay with a curiosity attitude, but also with an attitude “I do not accept artificial meat as an alternative to conventional meat, I eat meat substitutes though”. Extending these results to the huge Chinese population of China, we can speculate that most people may have a wait-and-see attitude towards artificial meat based on the uncertainty of this novel product, which is an obstacle needed to be resolved by the proponents of artificial meat.

#### 4.3.3. Reasons for Willingness to Try

The most important reasons to try artificial meat mainly focus on food production and safety issues to satisfy the nutritional needs of the Chinese population. Indeed, in China, saving food has always been a topic of great concern to society especially after the great Chinese famine (1958–1962) and the conception of saving food is deeply rooted in the hearts of the Chinese people. Therefore, it is highly inadvisable to waste food under the premise that there are 815 million people in a state of food shortage today [52]. Artificial meat is expected to produce meat in a highly effective way to solve the estimated increase in meat demand [53]. Chinese people would be willing to make a contribution to the world hunger issues by eating artificial meat.

A quarter of respondents had a willingness to try by curiosity. Since with the constant update of novel concepts and the rapidly increase of middle-class in China, people are looking for luxury food with a high curiosity and acceptance of new products [6]. Although it was also observed that the consciousness about animal welfare is growing as Chinese incomes and the industrialization of animal farming increases [54], less than a quarter of people aimed to improve animal welfare and reduce suffering through artificial meat. Among all the options, environmental- and ethical-based considerations ranked quite behind, indicating that Chinese consumers would be willing to try artificial meat not mainly due to the concerns of ethics and environment. In addition, one possibility which cannot be ruled out is that people still have not understood the potential advantages of artificial meat for the environment and ethics compared to conventional meat. However, we may conclude here that, despite the fact that the environmental issues have attached worldwide attention, Chinese people still seem not to pay much attention to these concerns compared to other issues such as food hunger in particular with the artificial meat topic.

#### 4.3.4. Reasons for No Willingness to Try

Previous observations indicated that in developing countries, concerns about environmental animal welfare and food security issues is less than in developed countries [40,55]. This may be true concerning environment and animal welfare issues which might be not as great as in Western countries. However, given the particular case in China, people worry a lot about food safety, which is indeed a major global public health concern and even particularly serious in China. Major sources of food safety issues in China are illegal additives and chemical contamination [56]. In China, the “green food” label—an eco-certification scheme for food—is used to certifies both the production process and the product characteristics [57]. With this organic label, natural, safety, high-quality, and nutritious properties of food will be guaranteed and trusted by Chinese consumers subsequently with a higher willingness to pay [58]. Artificiality, which has a negative connotation of unnatural, chemical, and lab-produced, contrasts with the principles of green food. This will negatively influence Chinese consumers’ attitudes toward artificial meat to a large extent. Although the term “artificial meat” is likely to highlight the unnaturalness of the product, the use of other terms such as “cultured meat” still cannot get rid of the fact that this novel product is not produced naturally. In fact, the safety issue is also a concern of some scientists, who claimed that the process of artificial meat production is never perfectly controlled, and some unexpected biological mechanisms could occur with potential negative impacts on human health or for the environment [59].

Consistent with the previous literature, unnaturalness was cited as one of the biggest barriers to consumer acceptance that could induce emotional resistance and lead to rejection of the novel product [48,60,61]. On the opposite side, Anderson and Bryant [62] underlined that naturalness does not always mean goodness, whereas unnaturalness does not always mean badness. But according to Deckers [63], naturalness is a deeply rooted worldview, the current development of artificial meat is challenging this worldview that is unlikely to be an effective advantage for persuading consumers [61]. According to our observations, concerns regarding safety and unnaturalness might be the most barriers for Chinese people’s willingness to engage with artificial meat. In addition, although a greater preference for artificial meat is significantly consistent with a lower price [49], the high price is not the main reason for Chinese people to reject the product at this stage, probably because respondents are not aware of the real price in practice since this novel product is not currently available commercially [8].

#### 4.3.5. Expectations for Artificial Meat

In line with the previous findings, safety is the most critical expectation for artificial meat. Interestingly, what really matters for Chinese people, in addition to safety, are the instinctive expectations for food mainly taste and nutritional value. The expectations about environment protection, price, and animal welfare were less important than safety, taste, and nutritional value in the Chinese context. As a consequence, to improve the acceptance of this novel food by the Chinese consumer, research and development should be driven by improvements in safety, taste, and nutrition value.

#### 4.3.6. The Potential Possibility to Engage with Artificial Meat

Dempsey and Bryant [19] interviewed stakeholders in China who acknowledged that the barriers for artificial meat in the Chinese market will differ from those in West countries. Indeed, Chinese respondents who mainly focus on safety, nutritional value, origin, and traceability during their food purchase were more unlikely to try artificial meat compared to respondents who mainly focus on ethics, energy intake as well as marks and labels who were more willing to try artificial meat. Safety, nutritional value, and traceability of food are shown to be the main drivers for greater profitability [64]. Thus, guaranteeing and improving factors that matter for Chinese consumers can improve the adoption of this novel food.

Previous studies indicated that most consumers are willing to try artificial meat, but a relatively small proportion would choose it over conventional meat or other meat alternatives [8]. In our study, nearly half of the respondents would be willing to try (29.8% probably, 19.9% definitely), close to half of the respondents refuse to eat artificial meat regularly, the rest of the others would be willing to eat regularly depending on conditionals (around one third prefer at a restaurant, at home, and ready-to-eat meals, respectively), nearly both half of them would be willing to accept artificial meat as an alternative to conventional meat. In addition, 10.7% of the respondents who would try artificial meat and 14.4% of the respondents who would eat artificial meat regularly would accept artificial meat instead of meat substitutes. By contrast, a third of the respondents would try artificial meat and eat meat substitutes, but they do not accept artificial meat as an alternative in the future. China has a long history of producing and eating meat substitutes made with soybean proteins. In the light of the present results, Chinese consumers would still prefer to consume plant-based meat substitutes rather than accepting the idea that artificial meat might become a viable alternative in the future.

Food neophobia and food disgust have significant effects on the willingness to consume artificial meat [50]. We observed that once people have emotional resistance and negative feeling such as absurdity and/or disgusting, they would be likely unwilling to try and to eat regularly artificial meat. On the other hand, for the participants who had no willingness to eat regularly, half of them thought artificial meat is “fun and/or intriguing”. However, among people having a positive perspective such as “promising and/or acceptable”, one-third of them had no willingness to eat it regularly yet, which indicates that being very positive toward artificial meat does not induce any great willingness to eat it regularly.

Wilks and Phillips [65] reported that 65% would be willing to try artificial meat, but among them, 33% would be willing to eat it regularly. Weinrich et al. [16] found that 57% of German consumers are willing to try artificial meat and only half of those would be willing to eat regularly. More positively, in the current study, 20% of respondents would be definitely willing to try and among them, 64% would be willing to eat regularly. In the same way, 30% would be probably willing to try and, among them, 63% would be willing to eat regularly. In brief, being willing to try artificial meat does not mean people would be willing to eat regularly, which is understandable. What is unanticipated is that a few proportions of respondents indicated that they were willing to eat artificial meat at a restaurant, home-made, or in ready-eat-meals although, they had even no willingness to try artificial meat. Indeed, we could have speculated that, if people refuse to try at the first step, they would be reluctant to engage further with artificial meat. Alternatively, it can be interpreted as a strong and direct wish of people to eat regularly artificial meat, with no need of a first try, or without any need of previous or specific information about the type of meat in dishes. Another possibility can be that when some specific options of food (burger, lasagna, or eating at a restaurant) were offered to consumers, it will stimulate their subconscious willingness through thoughts such as “it’s OK to eat” or “why not eat?” This observation reveals that it is difficult and tricky to predict consumers’ willingness in practice. Respondents’ answers might be subject to bias because indicating our own attitudes and willingness to engage with a product which is not yet available is difficult [8]. Sharma et al. [66] demonstrated that consumer acceptance could be the biggest barrier that artificial meat faces, exploring the solution to address the above difficulties is a long way off.

### 4.4. Future Development

The current respondents were asked about the potential future development and availability of artificial meat. Some producers are expected that artificial meat will be available in markets within five years [67]. Moreover, recent forecasting reports have estimated by the year 2040, 35% of the global meat production may be cultivated [68]. The most optimistic artificial meat companies currently estimate that artificial meat will be commercially available by 2021 [69]. Furthermore, Mark Post, who created the world’s first artificial meat hamburger, also claimed lately that cultured meat will become a food commodity in the near future due to the technological advances and investment in cultured meat [70].

For the current Chinese respondents, most people estimated that artificial meat will become realistic in their daily life, a small proportion (10.3%) of people expressed their strong conviction that artificial meat will never come to the realistic world. Taken into account the centralized policy and regulatory environment in China, the development of the artificial meat industry in China would be heavily dependent on governmental decisions, the latter being partially determined by levels of acceptance amongst the general public [19]. With the fact that artificial meat became authorized for sale in late 2020 in Singapore (the first nuggets with artificial meat are available in a restaurant) [71], it is questioned if China will follow its Asian neighbor.

In answer to our questions concerning the public and private research models, less than half of respondents trust the public research or private research for developing research on artificial meat, which implies that more than half have no trust either on public research or on the private model. The aforementioned fact that the majority of respondents expressed their attitudes by “unsure” and “probably” indicates that a large part of respondents was actually swinging between acceptance and rejection. However, for Chinese consumers, public research centers or government-related organizations are more reliable than private companies such as startups. The Chinese may have less willingness to purchase products from startups, even if that private companies are likely to receive public funding. By contrast, if the product is developed by governmental organizations, Chinese people may have more willingness to try, to eat, and to pay for it. After all, there is a gap between expectations and reality, since there is no yet commercialized artificial meat to more precisely evaluate consumers’ behavior.

## 5. Conclusions

This study investigated consumers’ attitudes, perspective, willingness, and potential acceptance of artificial meat in China. According to the current responses, nearly half of the respondents would be willing to try artificial meat, close to half of the respondents refuse to eat artificial meat on a regular basis, the rest would be willing to eat regularly depending on conditions. However, new information provided by this study is that nearly 90% of the respondents would be willing to pay less for artificial meat compared to conventional meat. This observation is important following the fact that this novel product has been recently approved for sale in Singapore. Most people agree with the weaknesses of current conventional meat production and therefore may consider the potential advantages of artificial meat with respect to environmental, ethical, and health issues. These views are confirmed by the fact that a significant part of respondents would be naturally willing to reduce their meat consumption and to try and eat artificial meat as a meat substitute. For Chinese consumers, safety is the most important issue. In this context, perceived unnaturalness of artificial meat can induce a feeling of insecurity or emotional resistance and, thereby further lead to the rejection of this novel product with no willingness to eat regularly, which would be a difficult obstacle to overcome. Unlike in Western countries, ethics and environmental issues are not the main drivers of acceptance of artificial meat for Chinese consumers at this stage. In addition, this novel food is expected to be safe, tasty, and nutritious at least as much as the conventional meat for the current consumers. According to the present results, it would make sense to focus on important issues for Chinese consumers to promote any type of food products on the basis of Chinese catering culture, perspective on food, and traditional philosophy. Otherwise, anticipation of the success of artificial meat in China should be less optimistic.

## Figures and Tables

**Figure 1 foods-10-00353-f001:**
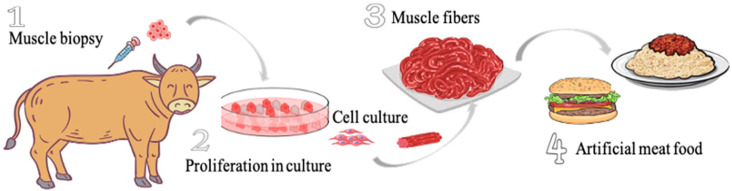
Briefing provided to respondents regarding the introduction of artificial meat.

**Figure 2 foods-10-00353-f002:**
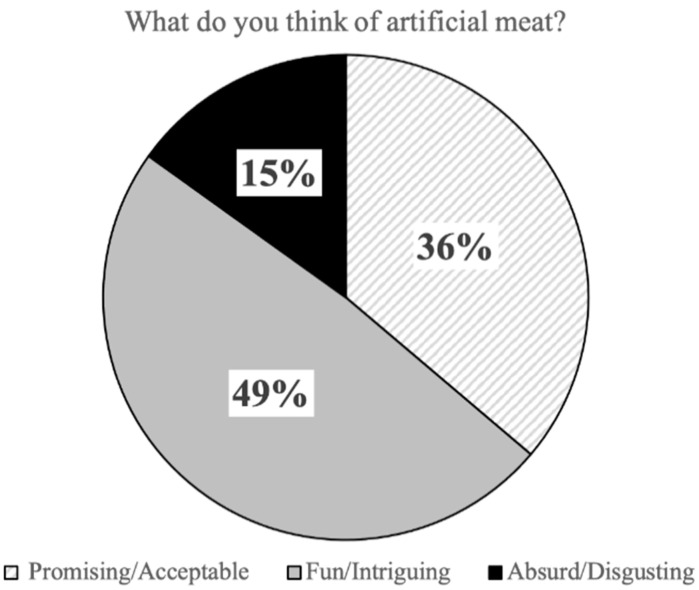
General opinion of the respondents about artificial meat.

**Figure 3 foods-10-00353-f003:**
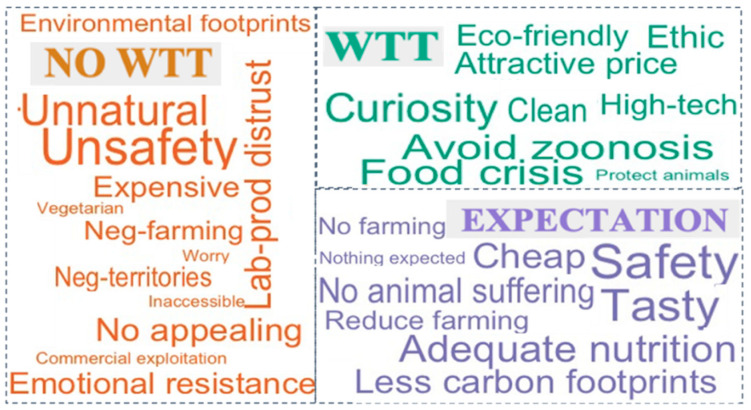
Word cloud plot on the reasons for willingness to try (WTT), no willingness to try (NO WTT) and the expectations from artificial meat.

**Figure 4 foods-10-00353-f004:**
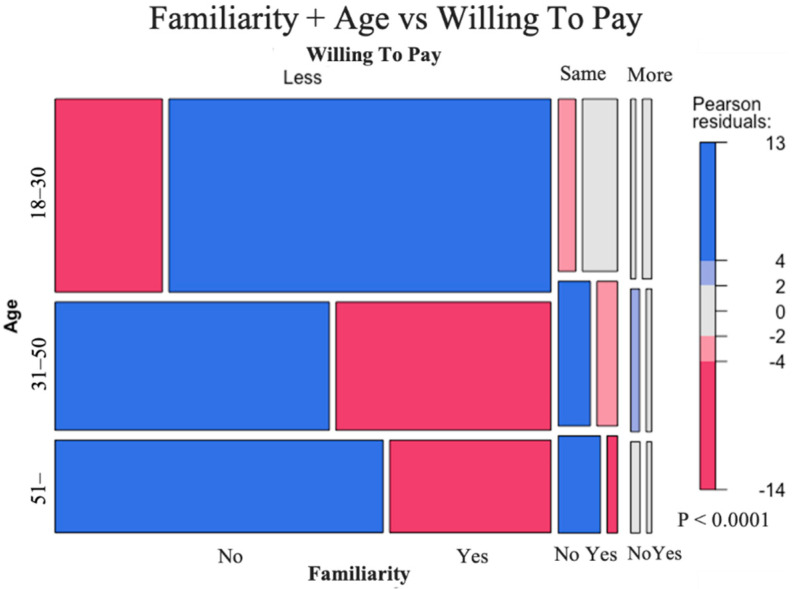
Mosaic plot of the distribution of responses about willingness to pay (WTP) according to age and familiarity.

**Figure 5 foods-10-00353-f005:**
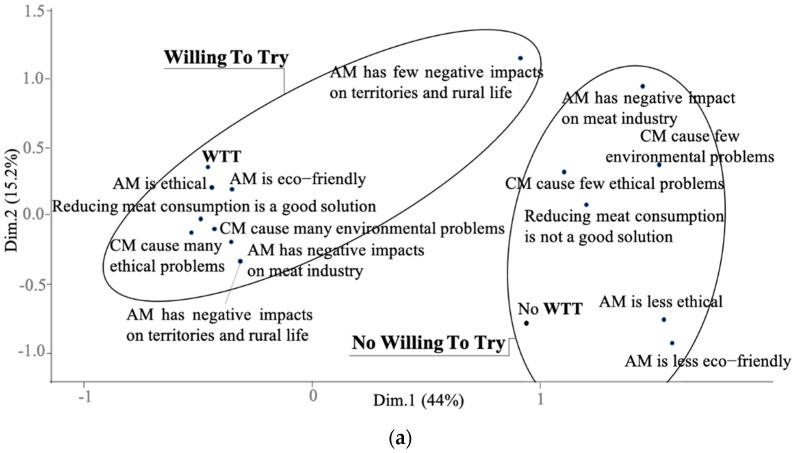
Multiple components analysis: (**a**) the relationships between societal challenges and willingness to try (WTT) artificial meat; (**b**) the relationships between societal challenges and willingness to eat (WTE) artificial meat. WTE: willing to eat regularly; No WTE: no willing to eat regularly; WTT: willing to try; No WTT: no willing to try; AM: artificial meat; CM: conventional meat.

**Figure 6 foods-10-00353-f006:**
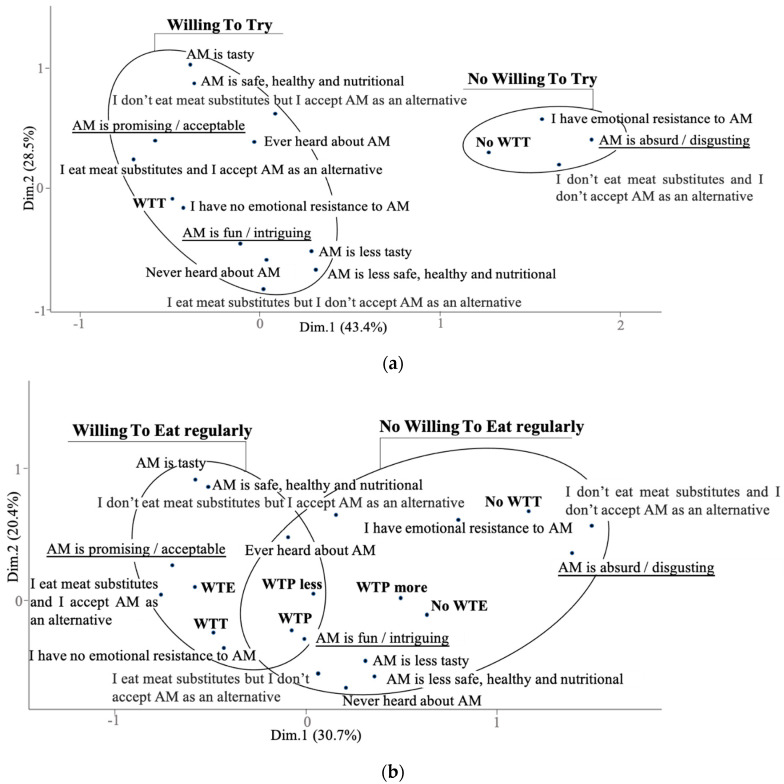
Multiple components analysis: (**a**) the relationships between perceptions about artificial meat characteristics and willingness to try (WTT); (**b**) the relationships between perceptions about artificial meat characteristics and willingness to eat (WTE) artificial meat. WTE: willing to eat regularly; No WTE: no willing to eat regularly; WTT: willing to try; No WTT: no willing to try; WTP: willing to pay the same as conventional meat; WTP more: willing to pay more than conventional meat; WTP less: willing to pay less than conventional meat; AM: artificial meat; CM: conventional meat.

**Figure 7 foods-10-00353-f007:**
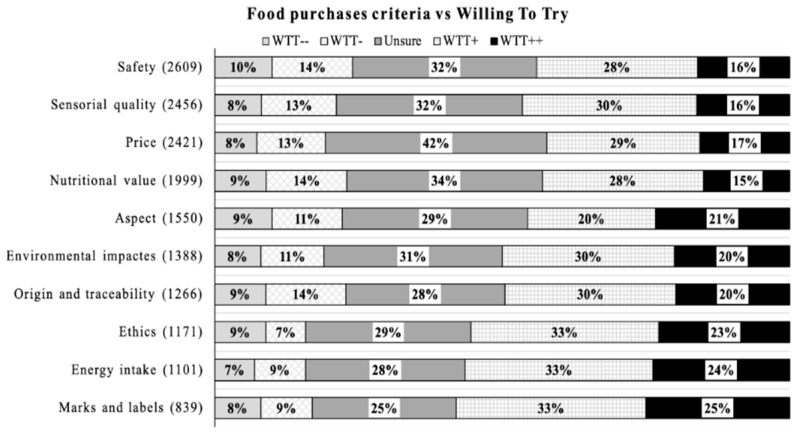
Cross analysis between food purchase criteria and respondents’ willingness to try (WTT). Note: the number in the parentheses is the number of respondents who choose the corresponding option. WTT: willing to try; WTT−−: definitely no WTT; WTT−: probably no WTT; Unsure: unsure to try; WTT+: probably WTT; WTT++: definitely WTT.

**Figure 8 foods-10-00353-f008:**
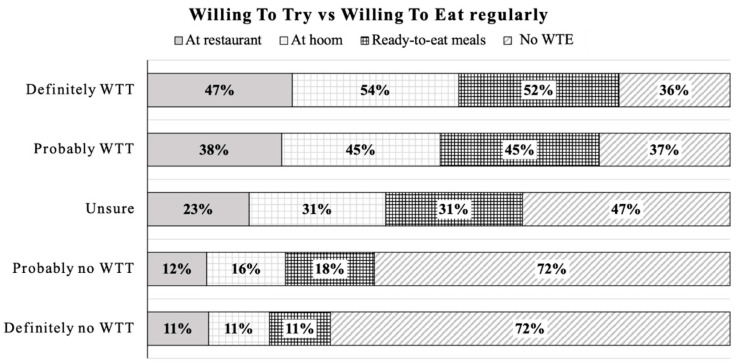
Cross analysis between willingness to try (WTT) and willingness to eat regularly (WTE). WTT: willing to try; No WTE: no willing to eat regularly.

**Figure 9 foods-10-00353-f009:**
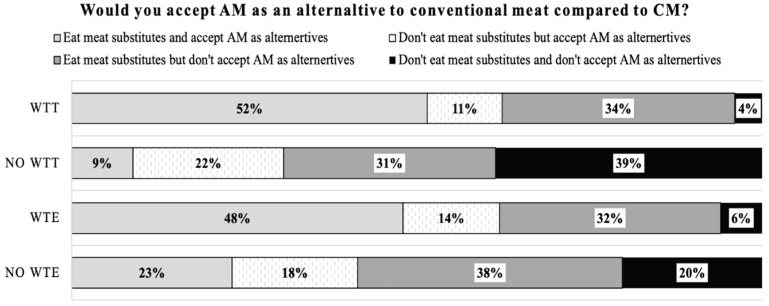
Cross analysis between willingness to try (WTT), willingness to eat regularly (WTE) and accepting artificial meat as an alternative. WTT: willing to try; No WTT: no willing to try; WTE: willing to eat regularly; No WTE: no willing to eat regularly.

**Figure 10 foods-10-00353-f010:**
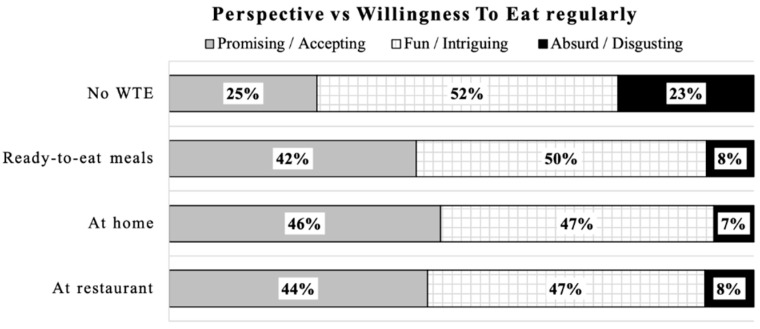
Cross analysis between respondents’ perspective and willingness to eat regularly (WTE). No WTE: no willing to eat regularly.

**Table 1 foods-10-00353-t001:** Demographic information of the respondents of our survey (4666 responses).

Question	Response Option	No. ^1^	% ^2^	Mean	SD
Sex	No wish to answer = 0	148	3.17	1.43	0.56
	Female = 1	2360	50.6		
	Male = 2	2158	46.2		
Age	18–30 years of age = 1	2149	46.1	1.76	0.79
	31–50 years of age = 2	1470	31.5		
	>51 years of age = 3	1047	22.4		
Education	No wish to answer = 0	141	3.02	2.12	0.84
	Primary/High/Engineer school = 1	695	14.9		
	College = 2	2596	55.6		
	Master = 3	951	20.4		
	PhD = 4	283	6.07		
Activity area	Meat scientist = 1	276	5.92	3.46	0.96
	Other scientist = 2	760	16.3		
	Meat worker = 3	164	3.51		
	Others = 4	3466	74.3		
Monthly net income	No wish to answer = 0	1026	21.9	2.35	1.82
	<1 k yuan (155 USD) ^3^ = 1	816	17.5		
	1 k–3 k (155–464 USD) = 2	549	11.8		
	3 k–5 k (464–773 USD) = 3	834	17.9		
	5 k–8 k (773–1236 USD) = 4	945	20.2		
	8 k–10 k (1236–1546 USD) = 5	210	4.50		
	>10 k (1546 USD) = 6	286	6.13		
Meat consumption	Never: vegetarian or vegan diet = 1	157	3.36	3.08	0.86
	Rarely: weekly or less = 2	1099	23.6		
	Regularly: several times a week = 3	1631	34.9		
	Daily or within each meal = 4	1779	38.1		
Familiarity ^4^	Yes = 1	2611	55.9	1.44	0.50
	No = 2	2055	44.1		

^1^ No.: number of responses; ^2^ %: percentage of responses; ^3^ yuan: Chinese yuan renminbi (CNY), 1 CNY = 0.15 USD (American dollar); 1 CNY = 0.13 Euros, on 21 January 2021; ^4^ Familiarity: have you heard about artificial meat?

**Table 2 foods-10-00353-t002:** Responses about societal challenges related to the meat industry and the perceptions about artificial meat (4666 responses).

**A. Question–Societal Challenges**	**Response (1 Much Less–5 Much More)**						
	1 ^1^	2 ^2^	3 ^3^	4 ^4^	5 ^5^	Mean	SD
Do you think the conventional meat industry causes ethical problems?	748 ^6^ (16.0) ^7^	594 (12.7)	1181 (25.3)	1110 (23.6)	1033 (23.8)	3.23	1.36

Do you think the conventional meat industry causes environmental problems?	385 (8.3)	479 (10.3)	1469 (31.5)	1337 (28.7)	996 (21.3)	3.45	1.17

Do you think reducing meat consumption could be a good solution to resolve above problems?	551 (11.8)	646 (13.8)	1089 (23.3)	1377 (29.5)	1003 (21.5)	3.35	1.28

How ethical do you think artificial meat would be compared to conventional meat?	431 (9.2)	453 (9.7)	1567 (33.6)	1292 (27.7)	923 (19.8)	3.39	1.18

How eco-friendly do you think artificial meat would be compared to conventional meat?	324 (6.9)	355 (7.6)	1496 (32.1)	1517 (32.5)	974 (20.9)	3.53	1.11

Do you think artificial meat have negative impacts on conventional meat industry?	290 (6.2)	411 (8.8)	1476 (31.6)	1480 (31.7)	1009 (21.6)	3.54	1.11

Do you think artificial meat have negative impacts onterritories and rural life?	348 (7.5)	571 (12.2)	1338 (28.7)	1358 (29.1)	1051 (22.5)	3.47	1.18

**B. Question–Perception**	**Response (1 Much Less–5 Much More)**						
	1	2	3	4	5	Mean	SD
How healthy, safe and nutritional do you think artificial meat would be compared to conventional meat?	1098	1358	1614	409	187	2.41	1.06
	(23.5)	(29.1)	(34.5)	(8.7)	(4.0)		
How tasty do you think artificial meat would be compared to conventional meat?	1279	1556	1292	358	181	2.27	1.06
	(27.4)	(33.3)	(27.7)	(7.7)	(3.9)		
Do you have emotional resistance to try artificial meat?	1130	1311	1472	412	341	2.47	1.16
	(24.2)	(28.1)	(31.5)	(8.8)	(7.3)		

^1^ 1 Much less, ^2^ 2 Less, much less-1 and less-2 sometimes were analyzed together; ^3^ 3 Neutral; ^4^ 4 More, ^5^ 5 Much more, more-4 and much more-5 sometimes were analyzed together; ^6^ Number of responses; ^7^ (percentage of responses).

**Table 3 foods-10-00353-t003:** Responses about willingness to engage with artificial meat and the future development.

**A. Question–Willingness to Try, to Eat Regularly and to Pay for Artificial Meat**	**No.**	**%**	**Mean**	**SD**
Would you be willing to try artificial meat?				
1 Definitely no	446	9.6	3.39	1.20
2 Probably no	531	11.3		
3 Unsure	1369	29.3		
4 Probably yes	1390	29.8		
5 Definitely yes	930	19.9		
Would you accept artificial meat as an alternative to conventional meat compared to meat substitutes?				
1 Yes and I eat meat substitutes	1681	36.0	2.24	1.08
2 Yes but I don’t eat meat substitutes	791	16.9		
3 No but I eat meat substitutes	1593	34.1		
4 No and I don’t eat meat substitutes	601	12.9		
In which context(s), would you be willing to eat artificial meat regularly? (multiple choice)				
At the restaurant	1447	29.7		
At home	1731	35.5	-	-
In ready-to-eat meals: lasagna, burger…	1734	35.6		
I do not want to eat artificial meat regularly	2299	47.2		
How much would you be willing to pay for artificial meat compared to conventional meat?				
1 Much less than conventional meat, even nothing at all	2159	46.3	1.73	0.85
2 Less than conventional meat	1861	39.9		
3 Same price as conventional meat	478	10.2		
4 More than conventional meat	86	1.8		
5 Much more than conventional meat	82	1.8		
**B. Question–Future Development**	**No.**	**%**	**Mean**	**SD**
How long do you think artificial meat will become realistic				
1 In the short term: 1–5 years	1021	20.9		
2 In the medium term: 6–15 years	2194	45.1	2.24	0.89
3 In the long term: >16 years	1152	23.7		
4 Never	501	10.3		
Do you think that public research must invest (time and funding) to develop this biotechnology?				
1 Much less	1179	6.9		
2 Less	1345	7.6		
3 Unsure	1630	39.5	2.42	1.11
4 More	454	29.2		
5 Much more	260	16.8		
Do you think a private research model (start-ups) is relevant for potentially developing research on artificial meat?				
1 Much less	334	24.2		
2 Less	370	27.6		
3 Unsure	1925	33.5	3.43	1.07
4 More	1421	9.3		
5 Much more	818	5.3		

**Table 4 foods-10-00353-t004:** Analysis of variance of willingness to try according to demographics.

Willing to Try	*p* Value	Willing to Try	*p* Value
Sex	0.000	Sex × Age	0.000
Age	0.000	Sex × Education	0.000
Education	0.000	Sex × Activity	0.004
Activity area	0.000	Sex × Income	0.04
Income	0.000	Sex × Meat Consumption	0.049
Meat consumption	0.000	Age × Education	0.000
Familiarity	0.000	Age × Meat Consumption	0.000

The mean difference is significant at the 0.05 level. Only significant interactions are indicated.

**Table 5 foods-10-00353-t005:** Pairwise comparisons between significant demographic groups for willingness to try.

	**A-Age**				**B-Education**				**C-Activity Area**				
Sex	18–30	31–50		>51	P/H/E ^1^	University	Master	PhD	MeatS ^2^	Meat W ^3^		Other S ^4^	Others ^5^
Female	3.05 ^d^	3.41 ^c^		3.69 ^b^	3.3 ^c,d^	3.16 ^d^	3.72 ^b^	3.31 ^c^	3.81 ^a,b^	4.16 ^a^		3.31 ^c^	3.18 ^c^
Male	3.08 ^d^	3.87 ^a^		3.89 ^a^	3.41 ^b^	3.47 ^b^	3.91 ^a^	3.62 ^b,c^	3.83^ab^	3.57 ^b^		3.54 ^b^	3.56 ^b^
	**D-Income**								**E-Meat consumption**				
Sex	<1 k		1–3 k	3–5 k		5–7 k	8–10 k	>10 k	Never	Rarely	Regularly		Daily
Female	3.07 ^d^		3.37 ^c^	3.29 ^c,d^		3.69 ^b^	3.50 ^a,b,c^	3.72 ^a,b^	2.26 ^e^	3.02 ^d^	3.22 ^c^		3.68 ^b^
Male	3.12 ^d^		3.59 ^b,c^	3.81 ^a,b^		3.90 ^a^	3.81 ^a,b^	3.55 ^b^	1.86 ^e^	3.20 ^c^	3.55 ^b^		3.84 ^a^
	**F-Education**								**G-Meat consumption**				
Age	P/H/E			University		Master	PhD		Never	Rarely	Regularly		Daily
18–30	3.04 ^e^			2.99 ^e^		3.35 ^c,d^	3.39 ^c,d^		2.54 ^e,f^	2.95 ^e^	3.09 ^d,e^		3.15 ^d^
31–50	3.4 ^d^			3.66 ^c^		3.92 ^b^	3.42 ^c,d^		2.04 ^f^	3.09 ^d,e^	3.56 ^c^		4 ^b^
>51	3.39 ^d^			3.86 ^b^		4.15 ^a^	3.88 ^a,b,c^		1.72 ^f,g^	3.33 ^c^	3.83 ^b^		4.16 ^a^

^1^ P/H/E: primary school/high school/engineering school; ^2^ Meat S: meat scientist; ^3^ Meat W: meat worker; ^4^ Other S: scientist work outside meat sector; ^5^ Others: people not scientist and outside the meat sector; Based on estimated marginal means; ^a–e^: different letters in the same section indicate means with significant statistical differences at the 0.05 level; To correct for multiple testing, a Bonferroni-adjustment was used.

**Table 6 foods-10-00353-t006:** Analysis of variance of willingness to eat regularly/willingness to pay according to demographics.

**Willing to Eat**	***p* Value**	**Willing to Eat**	***p* Value**
Sex	0.026	Familiarity × Income	0.014
Age	0.292	Income × Age	0.005
Education	0.070	Income × Sex	0.042
Activity area	0.017		
Income	0.004		
Meat consumption	0.920		
Familiarity	0.000		
**Willing to Pay**	***p* Value**	**Willing to Pay**	***p* Value**
Sex	0.060	Familiarity × Age	0.005
Age	0.008	Age × Education	0.003
Education	0.150	Age × Meat consumption	0.008
Activity area	0.870	Sex × Meat consumption	0.024
Income	0.009		
Meat consumption	0.170		
Familiarity	0.190		

The mean difference is significant at the 0.05 level. Only significant interactions are indicated.

## Data Availability

The data presented in this study are available on request from the corresponding author. The data are not publicly available because they concern consumers’ expression.

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
