# Peer review of "Chinese Consumers’ Attitudes and Potential Acceptance toward Artificial Meat"

_foods, 2021, doi:10.3390/foods10020353_

Round 1

Reviewer 1 Report

Lines

Comments

34

“Conventional meat production systems are”

44

Those expectations are hypothetical. It is important to pass to the reader a realistic message about that. The way is written it seems that it is a true hypothesis. Although there is a significant hype about the sector, the artificial meat industry still needs to get established as a solid and consistent segment to be considered a game changer. Please rewrite.

48-49

This affirmation about western countries is based on what? Any modelling data? Mancini’s and Bryant’s papers are about acceptance, as well as Shaw’s, and Weinrich’s. Authors need to be careful when presenting this type of information. This proposed solution (addressing problems of meat production in the West) must have supporting prediction data instead of not confirmed hypothesis.

65-66

What is the definition of clean?  Using less resources make the product more sustainable, not necessarily clean. Please clarify. The word clean is usually associated to wholesomeness and hormone-free

69-71

This statement directly contradicts previous affirmations on lines 44-49. That’s why is important to make sure that expectations are still hypothetical.

83

Define ESOMAR

96

Is this number from Brazil or China?

174

Please present this value in USD and €, add € on the table footnote and the date that it was quoted. (line 181)

837-844 (based on 268)

It is really interesting that the studied group is prone to try and accept artificial meat and support public research. However, it is known that startup companies receive public funding but it seems that a smaller amount of the respondents, when compared to the ones who support public research, didn’t support start up model. Can this be an indication that the population in general does not understand how startups work?

Author Response

Dear Editor

Thanks very much for the review of our manuscript and the valuable comments of the referee.

Please, see the attachment with answers to these comments and a revised version of our manuscript (modifications in red) considering these comments.

Regards

J Liu and JF Hocquette

Reviewer 2 Report

The manuscript entitled “Chinese consumers’ attitudes and potential acceptance toward artificial meat” presents an interesting issues associated with the artificial meat.

Introduction:

  • In this section Authors presented the information associated with the conventional meat production system and artificial meat. This section should be briefly presented – what do we know and what is the background for this study. As authors stated “Up to the present time, there have been various surveys regarding consumer acceptance of artificial meat” so some detailed information about other studies are necessary. The good background should present the history of problem, the current knowledge and scientific "gap", and then authors should present how their study could fill this gap to justify the study.
  • Authors should also emphasize the novelty of the study. The information that current study is conducted on “large group of Chinese consumers” is not sufficient.

Materials and Methods

  • Line 98 – please specify “avoid irrelevant answers to some extent”
  • Please provide the sample size calculation – and how it could be representative for socio-demographic various   
  • Please provide inclusion and exclusion criteria for respondents (e.g. vegetarian, etc.). This is crucial for the quality of the response and inferences (conclusion)
  • Line 122-124 – Was the normality of distribution tested? The information about it should be added and authors should be consequent. If data have normal distribution, they should be treated as such, if not, nonparametric tests should be applied. Please specify it.
  •  

Results and discussion

  • Please provide relevant psychometrics (due to the fact, that authors analyses the consumers’ attitudes)
  • Line 174 – please present values also in dollars or euros (for international readers) (not only that 1 CNY = 0.147 USD)
  • Figure 5 should be improved (maybe less text). In the present form is difficult to follow.
  • Line 432 – “Generally speaking” – should be omitted.
  • Line 470 – “Are Chinese consumers eager for artificial meat?” – the title of sub-chapters should not be a question
  • Line 471-472 – “A news report entitled “In India and China, consumers are eager for lab-grown meat. 471 In the US? Not as much.” was published on vox media on March 5, 2019 [28].” Please avoid not peer-revied publications
  • The discussion session should be shorted.
  • Line 545 – “In addition, during the COVID-19” – thanking into account the fact, that survey was conducted during the COVID-19 pandemic – the obtained results should be discussed in this specific context.  

Conclusion:

  • Authors should rewrite the conclusion to be more focus on the study rather than present general remarks.
  • The sub-section “Author Contributions:” should be specified in details.

Author Response

(The authors gave the same response as above.)

Round 2

Reviewer 2 Report

I appreciate the great efforts that the authors have made in response to my questions and concerns. However, there are some small issues that should be corrected:

  • Table 4 – it should be “<0.001” instead of “0.000”
  • Table 6 – it should be “<0.001” instead of “0.000”